# Left Ventricle Phenotyping Utilizing Tissue Doppler Imaging in Premature Infants with Varying Severity of Bronchopulmonary Dysplasia

**DOI:** 10.3390/jcm10102211

**Published:** 2021-05-20

**Authors:** Eunice Torres, Philip T. Levy, Afif El-Khuffash, Hongjie Gu, Aaron Hamvas, Gautam K. Singh

**Affiliations:** 1Department of Pediatrics, Washington University School of Medicine, St. Louis, MO 63110, USA; torres.eunice@wustl.edu; 2Division of Newborn Medicine, Boston Children’s Hospital and Harvard Medical School, Boston, MA 02115, USA; 3Department of Neonatology, The Rotunda Hospital, and Department of Paediatrics, The Royal College of Surgeons in Ireland, Dublin, Ireland; afifelkhuffash@rcsi.com; 4Division of Biostatistics, Washington University School of Medicine, St. Louis, MO 63110, USA; guh@wustl.edu; 5Department of Pediatrics, Northwestern University Feinberg School of Medicine, Chicago, IL 60611, USA; AHamvas@luriechildrens.org; 6Department of Pediatrics, Central Michigan University School of Medicine, Mt Pleasant, MI 48858, USA; GSingh3@dmc.org

**Keywords:** post systolic motion, left ventricular dysfunction, prematurity, tissue doppler imaging, echocardiography, bronchopulmonary dysplasia, pulmonary hypertension

## Abstract

Bronchopulmonary dysplasia (BPD) is characterized by alveolar-capillary simplification and is associated with pulmonary hypertension (PH) in preterm infants. The contribution of left ventricle (LV) disease towards this severe BPD-PH phenotype is not well established. We aimed to describe the longitudinal trajectory of the LV function as measured by tissue Doppler imaging (TDI) and its association with BPD-PH. We retrospectively assessed prospectively acquired clinical and echocardiographic data from 77 preterm infants born between 2011 and 2013. We characterized the LV function by measuring systolic and diastolic myocardial velocities (s’, e’, a’), isovolumetric relaxation time (IVRT), and myocardial performance index with TDI at three time periods from 32 and 36 weeks, postmenstrual age through one year of age. We also measured post systolic motion (PSM), a marker of myocardial dysfunction that results from asynchronous movement of the ventricular walls, and not previously described in preterm infants. Patients were stratified into groups according to BPD severity and the presence of PH and compared over time. Conventional TDI measures of the LV function were similar between groups, but the septal PSM was significantly prolonged over the first year of age in patients with BPD-PH. PSM provides a novel objective way to assess the hemodynamic impact of lung and pulmonary vascular disease severity on LV function in preterm infants with BPD and PH.

## 1. Introduction

Birth at extreme prematurity abruptly challenges the underdeveloped cardio–respiratory systems. The lungs must perform a gas exchange with a decreased surface area of both alveoli and pulmonary vasculature while the heart remodels due to the substantial hemodynamic changes associated with prematurity and its effects on its developmental programming [1,2,3]. The arrest of the alveolar-capillary unit following preterm birth leads to the most common respiratory complication in preterm infants, bronchopulmonary dysplasia (BPD) [4]. The original characterization of BPD included the findings of pulmonary vascular disease (PVD) and right ventricle (RV) hypertrophy [5]. It is now recognized that PVD, and its most severe form, pulmonary hypertension (PH), persist as major causes of morbidity and mortality of extremely premature infants [6].

While the prognosis of the BPD and PVD/PH is linked to RV function, the contribution of the immature left ventricle (LV) performance towards the BPD-PH phenotype is not as well characterized in this high-risk population. Premature birth limits the cell population available to support myocardial maturation and affects ventricular performance in preterm born infants [7]. Specifically, the terminal differentiation of myocytes that occurs in the third trimester dictates postnatal cardiomyocyte endowment, as the heart loses the proliferative capacity (hyperplasia) soon after birth [8]. With the increased recognition that preterm birth before 32 weeks of gestation is a risk factor for heart failure in childhood and young adulthood [9,10], there is a renewed interest in characterizing LV performance in preterm born individuals during critical windows of development over the first year of age.

Normal LV function requires coordinated sequential electrical activation and mechanical contraction patterns in the longitudinal, radial, and circumferential planes. Tissue Doppler imaging (TDI) derived post systolic motion (PSM), occurring during isovolumic relaxation, is a distinctive myocardial velocity pattern of LV longitudinal motion. This has been utilized in adults to detect serious myocardial dysfunction with abnormal and asynchronous movement of the ventricular walls that provides insight into both longitudinal contraction patterns and electrical stimulation [11,12]. TDI provides reliable parameters to assess the velocity of muscle tissue movement during systole and diastole in neonates [13], with important differences correlating to BPD severity [14,15,16]; however, the presence of LV abnormalities, as assessed by PSM, and its association to the development of the BPD-PH phenotype has not been characterized in neonates. 

Our group has previously demonstrated that BPD-PH adversely impacts contraction patterns in the circumferential–longitudinal plane (rotational mechanics) [17] with the preservation of LV longitudinal deformation (strain), [18] but there has not been any study to discern the presence of LV dysfunction using TDI derived PSM in preterm infants. We hypothesize that preterm infants with the BPD-PH phenotype have PSM as a marker of ventricular dysfunction and dyssynchrony over the first year of age. This is with the delineation of distinct velocity patterns during the time interval between the end of the LV systolic velocity wave and the onset of the early diastolic velocity wave that are not apparent with conventional TDI measures of the LV function. Accordingly, the objective of this study was to describe the trajectory of the LV function measured by TDI derived PSM in extreme preterm infants.

## 2. Experimental Section

### 2.1. Design and Population

For this exploratory study, we performed a post-hoc analysis on data from preterm infants enrolled through the Prematurity and Respiratory Outcomes Program (PROP, ClinicalTrials.gov: NCT01435187) site at Washington University School of Medicine/Saint Louis Children’s Hospital between August 2011 and November 2013. Details of the PROP study design have been comprehensively described elsewhere [19]. Briefly, 724 infants born between 230/7 and 286/7 weeks of gestation were recruited from six academic centers (13 hospitals) in the United States at birth and followed to one year corrected age (CA). Infants’ demographic and clinical characteristics were obtained at each site to identify early predictors of later respiratory morbidity [20]. At the Washington University PROP site, infants received echocardiograms at 32 weeks postmenstrual age (PMA), 36 weeks PMA, and one year CA. For this current study, 77 subjects were eligible for inclusion as they received echocardiograms at all three time points (Figure 1). Infants were excluded with any suspected congenital anomalies of airways, chest, and congenital heart disease (except for hemodynamically insignificant patent ductus arteriosus, ventricular septal defect, or atrial septal defect), chromosomal anomalies, neuromuscular disorders, fetal growth restriction, or were small for gestational age (birth weight < 10th percentile for gestation). A cohort of 50 healthy, age- and sex-matched term born individuals was recruited for the control group. These infants were referred to the Saint Louis Children’s Hospital Outpatient Pediatric Cardiology Clinic from August 2011 to November 2013 for heart murmur evaluation and found to have normal cardiac structure and function. The study was approved by the Institutional Review Board at Washington University. Written informed consent was obtained from the parents or guardians of participants.

### 2.2. Patient Characteristics

Demographic characteristics were collected at 32 weeks PMA, 36 weeks PMA, and one year CA. These time periods were chosen to match the acquisition timing of the echocardiograms according to the original PROP study design [18]. Physiologic and clinical biomarkers were also obtained at these time points, including heart rate, blood pressure, respiratory rate, level of supplemental oxygen, and respiratory support. The antenatal, delivery, and demographic characteristics were obtained. The following common neonatal outcomes were also evaluated: intraventricular hemorrhage (classified according to Papile Classification) [21], necrotizing enterocolitis (Bells stage II, with radiological evidence of pneumatosis), retinopathy of prematurity (stage 2 or higher), and presence of a patent ductus arteriosus (PDA) beyond 32 weeks PMA.

### 2.3. Primary Exposure of BPD-PH

We stratified infants into three groups according to BPD severity and the presence of pulmonary hypertension at 36 weeks PMA. BPD status was determined based on a modified version of the National Institute of Child Health and Human Development (NICHD) consensus definition [22] by utilizing the described respiratory support and/or oxygen requirement at 36 weeks postmenstrual age (PMA) [23]. Late PH was defined based on echocardiographic evidence of PH at 36 weeks PMA as having two or more of the broad- based criteria described by Mourani et al. [24] (and utilized in other studies [17,18,25]): estimated RV systolic pressure of more than 40 mmHg, a ratio of RV systolic pressure to systemic systolic blood pressure > 0.5, any cardiac shunt with bidirectional or right to left flow, RV hypertrophy or dilatation, or ventricular septal wall flattening. For this study, we defined Group I as infants with no or mild BPD; Group II as infants with moderate to severe BPD, and Group III as BPD of any grade complicated by late PH, labeled BPD-PH.

### 2.4. Echocardiography

Transthoracic echocardiograms were performed at three time points over the first year of age in the preterm infant: 32 and 36 weeks PMA, and one year CA. The timings of the echocardiograms were selected to avoid the early postnatal period of clinical and cardiopulmonary instability and early mortality associated with extreme preterm birth [17]. We performed echocardiograms in term controls at one year of age. Echocardiograms were obtained using a commercially available ultrasound imaging system (Vivid 7 and 9; General Electric Medical Systems, Milwaukee, WI, USA). One designated pediatric cardiac sonographer (T.S.) obtained all the echocardiographic images according to the American Society of Echocardiography guidelines [26], and one cardiologist provided the clinical interpretation. We employed neonatal protocols to acquire the images from the decubitus position during a restful period without changing the position of the infant or disturbing the hemodynamic condition to minimize heart rate and respiratory variation during the acquisition of the echocardiograms [18]. The image data were acquired digitally and stored in raw Digital Imaging and Communications in Medicine cine-loop format for offline analysis.

### 2.5. Tissue Doppler Imaging

We assessed the LV function offline from TDI derived longitudinal myocardial velocities from the apical four-chamber view. We measured the velocity of muscle movement of a single point along the myocardial wall at both the lateral and septal side of the mitral valve annuli from the base to apex in systole, and the reverse in diastole [13]. The longitudinal systolic, and early and late diastolic myocardial velocities of the LV lateral and septal walls were depicted as systolic velocity (s’), early diastolic velocity (e’, a measure of myocardial relaxation), and late diastolic (a’, a measure of late myocardial relaxation) waves (Figure 2), respectively. The e’/a’ ratio was calculated and used as a measure of diastolic function [27]. We assessed isovolumetric relaxation time (IVRT) as the time interval between the closure of the aortic valve and opening of the mitral valve occurring from the end of the s’ wave and the onset of the e’ wave, and isovolumetric contraction time (IVCT) as the time interval between the closure of the mitral valve and opening of the aortic valve occurring from the end of the a’ and onset of s’. We calculated the myocardial performance index (Tei index) [28], a validated measured of global ventricular function in neonates, from formula (a−b)/b [29]. In this formula “a” is equal to the sum of ejection time (ET), isovolumetric contraction time, and isovolumetric relaxation time, and “b” represents ET as displayed in Figure 2. We and others have previously demonstrated excellent feasibility and reproducibility for TDI derived longitudinal myocardial velocities [13].

### 2.6. Main Outcome: Post Systolic Motion (PSM)

We assessed PSM, also referred to as post systolic velocity, after termination of the s’ wave and prior to the onset of the e’ wave (Figure 3). PSM occurs in the same time interval where IVRT is measured. PSM is a distinctive velocity pattern that has been identified on left ventricular TDI measurements during a prolonged IVRT in the setting of myocardial dysfunction, that results in the abnormal and asynchronous systolic movement of the ventricular walls [11,12,30]. We measured PSM at all locations when present. All measurements were obtained by offline analysis and averaged from 3–5 cardiac cycles. Two observers performed a reproducibility analysis on 25 selected patients at all three time points using the Bland–Altman plot analysis (bias and limits of agreement), coefficient of variation, and intraclass correlation coefficient [18].

### 2.7. Statistical Analysis

Categorical level data are described as numbers and percentages, and continuous level data as medians and interquartile ranges. Normally distributed demographics, clinical parameters, and echocardiographic variables between groups were examined by a one-way analysis of variance (ANOVA), and group differences were compared through post-hoc testing using Tukey honestly significant difference testing. Non-normally distributed variables determined by the Shapiro-Wilk test were examined by chi-square, and independent samples by the Kruskal–Wallis one-way ANOVA. Relationships between outcome variables were examined using the univariate (Pearson product correlation coefficient), and standardized linear regression analyses with Bonferroni adjustment for multiple correlations were used to refine the ability to predict LV and septal PSM. The univariate analysis was used to determine the best predictors to enter in the model (significant correlation > 0.4), and then backward step-wise regression was performed to assess the independent effect of gestational age, gender, race, total oxygen days, length of stay, and common neonatal morbidities known to impact cardiopulmonary development (e.g., BPD and late PH) [17,18,25], while adjusting for weight and body surface area at each examination. Statistical significance was considered at *p* < 0.05. The independent effect of the risk factors for BPD and PH, including gestational age, gender, total oxygen days, length of hospital stay, perinatal factors, common neonatal morbidities (necrotizing enterocolitis, intraventricular hemorrhage, and retinopathy of prematurity), and presence of a patent ductus arteriosus (PDA) beyond 32 weeks PMA on PSM were also accounted for in the analysis. Due to the lack of data regarding the relationship between demographic and clinical characteristics with PSM, and the exploratory nature of this study, we used data from our previous work in premature infants and ventricular performance studies to estimate the sample size, assuming an alpha of 0.05, where 20 subjects per group would provide 99% power to detect 20% differences in echocardiographic measures between groups [17,18,25]. All statistical analyses were performed using SAS^®^ (SAS Institute Inc., Cary, NC, USA) 9.4 version.

## 3. Results

### 3.1. Patient Characteristics

Of 77 preterm born infants included in this study, 33 (43%) were classified in Group I, with no or mild BPD, 33 (43%) were in Group II with moderate or severe BPD, and 11 (14%) were in Group III with BPD-PH. Maternal and infant demographic and clinical characteristics for the preterm cohorts are summarized in Table 1. Infants in Group I were born at later gestational ages and had larger birth weights compared to Groups II and III (*p* < 0.01 for both). There were no differences in maternal complications between the three groups. Group I required fewer days of postnatal steroids, had less mechanical ventilation and oxygen therapy days, and shorter lengths of NICU (neonatal intensive care unit) stays compared to Groups II and III. Other than the difference in IVH (intraventricular hemorrhage) rate, the clinical data and postnatal complications were similar between Groups II and III. In this study, none of the preterm infants received inotropes, inhaled nitric oxide, or other pulmonary vasodilators at any time point in the first year of age. Additionally, none of the preterm born infants were on respiratory support at one year CA. Finally, all preterm born infants received caffeine from birth through 34 weeks PMA, and none received it beyond 34 weeks PMA. Of the 11 in the BPD-PH Group III, 4 (36%) had mild BPD, 5 (45%) had moderate BPD, and 2 (18%) had severe BPD (Table 2). None of the infants in this cohort had a cardiac catheterization and none had echocardiographic or clinical evidence of PH at one year CA.

### 3.2. Assessment of LV Tissue Doppler Indices

Comparisons of LV lateral and septal wall TDI measures between groups over time are presented in Table 3 and Table 4, respectively. There were no differences in e’, a’, s’, e’/a’, or MPI (myocardial performance index) between groups at 32 weeks PMA, 36 weeks PMA, or one year CA. All measures remained stable between 32 and 36 weeks PMA, even after adjustments for gestational age at birth, sex heart rate, and common co-morbidities. The following LV lateral and septal wall TDI measures increased from 36 weeks PMA to one year CA: e’, s’, and e’/a’ (*p* < 0.001 for all measures). LV lateral and septal wall a’ remained stable in all three groups from 32 weeks PMA to one year CA. LV lateral and septal wall MPI decreased from 36 weeks PMA to one year CA (*p* < 0.001 for measures). LV lateral IVRT was increased in Group III at 32 and 36 weeks PMA and was similar between all groups at one year CA. Septal IVRT was increased in Group III at all times points over the first year of age.

### 3.3. Assessment of Post Systolic Motion 

PSM was present on both the lateral and septal wall in 94% of the studies. LV lateral and septal wall PSM were similar between Groups I and II at 32 weeks PMA, 36 weeks PMA, and one year CA (*p* > 0.05 for all time points), and also remained stable from 32 weeks to one year CA (Table 3 and Table 4, Figure 4), even after accounting for gestational age and common neonatal morbidities. The LV lateral wall PSM was higher in Group III at 32 and 36 weeks PMA compared to Groups I and II (*p* < 0.01 for all), but similar between all three groups at one year CA; LV lateral wall PSM remained stable from 32 weeks to 36 weeks PMA, but significantly decreased from 36 weeks PMA to one year CA within Group III (*p* < 0.01) (Figure 4A). Septal wall PSM was higher in Group III at all time points compared to Groups I and II (*p* < 0.01 for all). PSM remained stable from 32 weeks to one year CA within Group III (Figure 4B). The same trends for lateral and septal PSM persisted when Group III was subdivided by the degree of BPD (no-mild (*n* = 4, 36%) and moderate-severe, (*n* = 7, 64%)) and compared to Group I and Group II, respectively. Within these 11 patients in Group III, there were no statistical differences between PSM based on the severity of the PH or the degree of BPD. Finally, there was significant correlation between IVRT and PSM for the lateral and septal walls (r = 0.67, *p* < 0.001). We observed high intra- and inter- observer reliability for both lateral and septal wall PSM (bias < 10%, narrow limits of agreement, ICC > 0.9, and coefficient of variation < 15% for each measure) (Table 5).

### 3.4. Comparison of Tissue Doppler Indices between Term and Preterm Cohorts

PSM was not present in the term infants. Septal wall e’, a’, s’, e’/a’, and MPI were similar between all preterm groups and term control infants at one year of age. Septal IVRT was the same between Groups I, II, and the term control population, but significantly higher when compared between Group III and the term controls (Table 5). LV e’ and s’ were higher (*p* < 0.01), and LV e’/a’, and MPI were lower in the term control cohort compared to all three preterm groups.

### 3.5. Influence of Patent Ductus Arteriosus

A patent ductus arteriosus (PDA) was present on the echocardiogram at 32 weeks PMA in 19% (*n* = 15) of infants with no difference between the three groups (Group I, *n* = 4, Group II, *n* = 9, Group III, *n* = 2). A PDA was present on the echocardiogram at 36 weeks PMA in 10% (*n* = 8) of infants with no difference between the three groups (Group I, *n* = 5, Group II, *n* = 1, Group III, *n* = 2). There were no PDAs present at one year CA. There were no differences in conventional TDI measures of PSM based on the presence of a PDA, adjusted for gestational age, sex, PDA size, diagnosis of BPD, and late PH (*p* > 0.05 for all measures at 32 and 36 weeks PMA, and one year CA). Similarly, there was no difference in measures at 32 and 36 weeks PMA or 1 year CA between groups, and those that required medical (19/77, 25%) and/or surgical intervention (9/77, 12%) to augment closure of the PDA. None of the infants had their PDA closed with a transcatheter device as this study was performed prior to the availability of the transcatheter device at our center.

## 4. Discussion

In this post-hoc analysis, we utilized tissue Doppler derived velocities of myocardial movement during systole and diastole to characterize the response of the LV in extreme preterm infants with varying degrees, and lung disease and PVD over the first year of age. For the first time to our knowledge in preterm born infants, we investigated TDI derived post systolic motion (PSM) as a marker of abnormal myocardial motion and synchrony and showed that those individuals with the severe BPD-PH phenotype had a significantly higher magnitude of septal wall PSM values that presented at 32 weeks PMA and persisted through one year of age. Interestingly, there were no differences over the first year of age in conventional systolic and diastolic tissue Doppler derived velocity measures based on the severity of BPD or presence of PH. The developmental patterns of PSM suggest that prematurity-related cardiorespiratory diseases impact LV myocardial function, reflected by abnormal longitudinal contraction patterns and asynchronous movement of the ventricular walls.

### 4.1. PSM in Neonates

Premature birth impacts LV development and growth, predisposing this population to long-term cardiovascular risk [9,10]. The LV consists of three fiber layers (superficial oblique, midwall circumferential, and deep longitudinal) that contribute to the complex movement of the LV with longitudinal translation, rotation and torsion, and thickening [31]. Several echocardiographic measures have been validated to separately assess longitudinal [18] and circumferential–longitudinal (rotational) [17] performance in preterm infants in both health and disease states [13]. However, a measure, such as PSM, that characterizes both the electrical and longitudinal contractile components and is influenced by the twisting of the myocardium may be a more sensitive measure of LV performance in neonates [11]. PSM has been shown to occur during prolonged IVRT, a period when part of the left myocardium untwists and changes the direction of movement [12]. The rate of untwisting depends on the energy generated from the systolic squeeze [32], and thus with the presence of PSM and prolonged IVRT there is a further reduction in the ventricular pressure required to open the mitral valve and generate suction to initiate ventricular filling. In our study, septal PSM was increased in BPD-PH infants, but e’ and a’ were similar amongst all preterm infants over the first year of age. In a small study of term neonates with PH, Patel et al. observed lateral tricuspid valve derived post systolic positive velocities occurring after the systolic s’ wave and before the e’, but they did not assess PSM on the left side [33]. The higher septal PSM likely reflects a maladaptive response of the electrical excitation of cardiac motion that begins in the base and travels to the apex in the BPD-PH infants. In this same cohort of preterm infants with abnormal cardiopulmonary disease, we previously demonstrated altered rotational mechanics with increasing predominance of basal clockwise rotation and decreasing predominance of the apical myofiber counterclockwise rotation [17]. These findings confirm the growing recognition that assessment of LV longitudinal performance in isolation may not detect subtle hemodynamic changes, as the LV longitudinal fibers occupy only 20% of the ventricular wall thickness and do not provide the main driving force on the LV to reduce ventricular diameter during contraction [34]. Previous studies that have assessed longitudinal function by conventional TDI [35] or deformation [18] also found no differences among preterm infants over the first year of age based on the severity of the cardiopulmonary disease [15,35].

In adults, PSM has been observed in situations created by ischemia or changes in loading conditions with abnormal motion of the myocardium secondary to ventricular asynchrony [11,12]. Although LV longitudinal mechanics are preserved in preterm infants and rotational mechanics demonstrate adaptive changes to different hemodynamic loading conditions during the late neonatal period with recovery by one year CA [17], altered RV performance manifested by increased RV afterload and RV dysfunction persists through one year of age, irrespective of the degree of lung disease or PVD [25]. Investigations in individuals born preterm have also shown that serial changes over time appear to be more adverse in the RV than the LV, with a reduction in function [36] and evidence of PVD/PH [37] compared to term born offspring throughout the ages [36]. The findings in our present and previous studies appear to support the recovery of the LV over time and persistence of RV dysfunction: (1) LV lateral PSM was higher at 32 and 36 weeks, but decreased by one year of age, mirroring the progression of LV torsion in neonates with prematurity-related cardiorespiratory complications; and (2) septal PSM remained abnormal through one year of age in infants with BPD-PH, reflecting the realization that the untwisting not only depends on the energy generated from the systolic squeeze, but also ventricular-to-ventricular interaction of the loading conditions [38]. The intrinsic characteristics of the preterm myocardium (e.g., immature contractile system with disordered myofibrils, an inefficient calcium handling system, and a non-compliant collagen) predispose the developing heart to poor tolerance to increased afterload, lack of reserve to cope with states of reduced preload, and diastolic dysfunction [27]. As a result, ventricles of BPD-PH premature infants become conditioned to an elevated afterload with a longer untwisting rate, relaxation time, and the presence of ventricular dyssynchrony, that is evident by the presence of prolonged septal PSM. 

### 4.2. Ventricular-Ventricular Interactions

The challenge in BPD-PH is for the RV to remain hemodynamically coupled to a compliant pulmonary arterial circulation. With the increased awareness of the impact that LV dysfunction has on the RV-PA interactions [39], it has been shown that prolonged LV IVRT also has a strong correlation with mean pulmonary artery pressure measured by echocardiography, and right heart catheterization in adult and pediatric patients with pulmonary hypertension [40]. When adaptive responses (e.g., hypertrophy and dilation) are no longer sufficient to compensate for an elevated RV afterload, the stretched myocytes with an increased sarcomere length will endure a sustained contraction that is less effective and conductive to dyssynchrony during ventricular relaxation [41]. A study of RV function in term neonates with PH attributed the post systolic velocity present in one-third of patients to sustained ventricular contraction after pulmonary valve closure [33]. We suspect that in our study the abnormal movement of the interventricular septum was the result of the pull felt by the septum when the prolonged contraction of the RV went unopposed by contraction in an LV myocardium that had already entered relaxation [42]. In support of this theory, others have shown that leftward septal motion has also been found to strongly correlate with multiple echocardiographic markers of LV diastolic dysfunction [38]. In addition to sharing the same oblique myocardial fiber, the RV and LV in the preterm infants both have diminished cardiomyocyte endowment, early loss of myocyte proliferative capacity [1], and decreased myocyte to extracellular matrix ratio [1,3] in the myocardium that may also explain the increase in PSM in the face of increased RV afterload over time.

### 4.3. TDI Measures at One Year of Age

There is an increasing body of literature on the use of TDI in preterm infants that focuses on diseases in the transitional period and the neonatal period, but with less emphasis beyond [13]. Poon et al. [43] was the first to track TDI measures in preterm infants from birth through one year of age and observed no difference in LV lateral wall e’, a’, or s’ between preterm and term individuals at one year. In contrast, compared to infants born at term, our study demonstrated lower LV lateral TDI measures in infants born preterm, irrespective of the degree of lung or pulmonary vascular disease. Our findings are in line with a recent meta-analysis, where Telles et al. demonstrated that infants born preterm compared to infants born at term have lower LV diastolic function (as reflected by lower e’ and a’ and higher e’/a’ ratio) and lower systolic function (as reflected by lower s’) during infancy, as well as across all developmental stages from childhood to adulthood [44]. We summarized the current literature that reports TDI measures at one year of age in term [43,45,46] and preterm TDI [43] infants in Table 6.

### 4.4. Clinical Significance of PSM Evaluation in Neonates

Given the clinical importance of earlier detection of PH in preterm infants with severe BPD [24], advances in risk assessment and monitoring for PVD, and right and left heart performance have led to more reliable quantitative modalities to detect and longitudinally follow the burden of disease and inform decision-making processes [47]. A preconditioned immature preterm myocardium responds poorly to changes in loading conditions, and reduced LV performance can exert significant load to the right ventricle (RV) that can affect RV-pulmonary vasculature (PV) coupling. The extent to which prematurity-related alterations in cardiopulmonary structures affect the complex LV performance has remained unclear due to the lack of early detection by a reliable biomarker of the disease, making the clinical management, intervention planning, and outcomes prediction for these patients challenging [13]. In this study, PSM correlated with IVRT, a recognized indicator of diastolic dysfunction with prolongation. IVRT could also be affected by electrical activity, such as delayed repolarization noted in cardiomyopathies and in patients with bundle branch block where IVRT may be prolonged. Prolongation of IVRT in these conditions reflects abnormal synchronous LV relaxation due to alterations in electrical activity, but these conditions also exhibit PSM due to abnormal rotational/torsional mechanics. Furthermore, IVRT and other conventional TDI measures focus on the longitudinal evaluation of LV performance, but PSM combines elements from the longitudinal, electrical, and rotational components of LV mechanics. PSM may provide a more comprehensive marker of LV performance (that was not previously available with conventional imaging or IVRT) to serially track cardiopulmonary disease over the first year of age. With an accepted physiological maturation pattern of PSM in preterm infants, we feel that this unique TDI derived parameter may provide a valid basis that allows comparison among studies and between health and disease [13]. 

### 4.5. Limitations

The implications of this study need to be understood within the context of its limitations. Although this is the first longitudinal study to track PSM in preterm infants, and only the second study to track TDI mechanics to one year of age in preterm infants, the small sample size and exploratory nature of the post-hoc study design limit the strong associations that can be made with this measure. The sample size calculation allowed for a univariate analysis to determine the best predictors to enter in the model and then backward step-wise regression to assess the independent effect of common perinatal confounders; however, casualty cannot necessarily be proven from the retrospective, nonrandomized design. None of the infants required pulmonary vasodilator therapy or inotropic support, suggesting that the PH in this cohort was relatively well controlled. Although the PSM findings may not be representative of critically ill neonates requiring more aggressive support for their PH or cardiovascular status, we can only hypothesize that the alterations with PSM and impact of LV performance may be even more pronounced with more severe phenotypes of PH. Future large-scale prospective studies are evolving to answer these questions [48]. We assessed the presence of a PDA on PSM between groups at 32 weeks PMA, 36 weeks PMA, and one year CA, but we did not account for clinically significant loading conditions on PSM related to atrial communication that modulate the effects of a presumed hemodynamically significant PDA [49]. Mechanical ventilation can also alter the loading conditions and impact PSM values. In this study, 6% (*n* = 2) of infants were on mechanical ventilation at 32 weeks PMA and 3% (*n* = 1) of the infants at 36 weeks PMA. There were no infants on respiratory support at one year CA. We were unable to properly investigate the effect of mechanical ventilation on PSM changes because all of the mechanically ventilated infants at 32 weeks and 36 weeks PMA developed BPD. Although several studies have explored RV TDI measures in BPD and PH patients [15,16,50,51,52,53], and one study reported RV PSM in neonate with PH [33], future work will focus on assessing PSM from an RV focused apical four-chamber view in preterm born individuals.

## 5. Conclusions

In this exploratory post-hoc analysis, we demonstrated for the first time in preterm born individuals that PSM is present and is a sensitive measure of abnormal LV performance in premature infants with varying degrees of cardiopulmonary disease in infants over the first year of age. Although conventional echocardiographic parameters have suggested that LV longitudinal function in premature infants has persevered, we have shown that LV function is indeed abnormal as demonstrated by the presence of PSM. PSM also correlated with IVRT on both the lateral and septal side of the LV and may provide further insight into recognized LV diastolic dysfunction in preterm infants. The presence of abnormal septal motion might also serve as a marker of the impact of disease severity on ventricular function in a population of infants with a high prevalence of cardio-pulmonary disease. The developmental patterns of PSM suggest that prematurity-related cardiorespiratory diseases impact myocardial function, reflected by abnormal longitudinal contraction patterns and asynchronous movement of the ventricular walls. Future prospective studies are now needed to track PSM throughout the ages in preterm born individuals to characterize its natural progression in both health and disease. Adequately powered prospective studies are required to better understand the role of post systolic motion as a marker of systolic and diastolic dysfunction in premature infants with different BPD phenotypes.

## Figures and Tables

**Figure 1 jcm-10-02211-f001:**
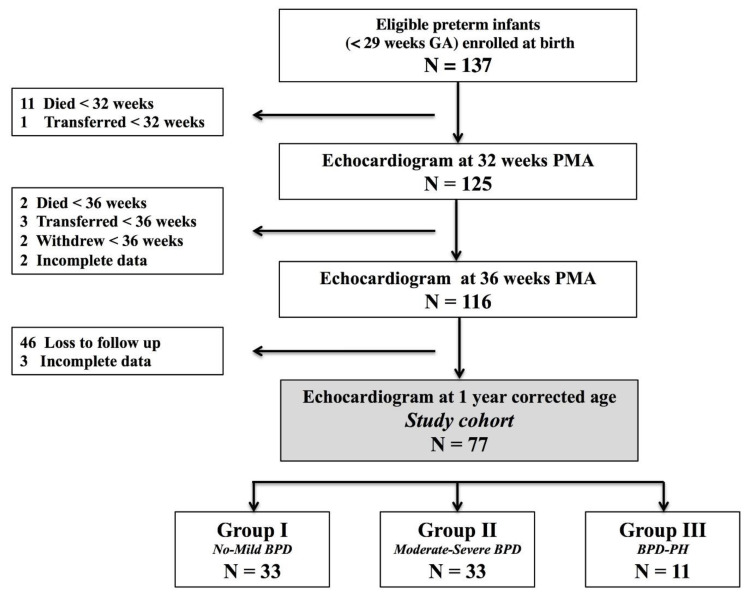
Study flow diagram of preterm infants. GA, gestational age. BPD, Bronchopulmonary dysplasia. PH, Pulmonary hypertension. Of the original 137 infants recruited for the Premature and Respiratory Outcomes Program (PROP) at birth, 77 had echocardiograms at 32 weeks postmenstrual age (PMA), 36 weeks PMA, and 1 year corrected age (CA).

**Figure 2 jcm-10-02211-f002:**
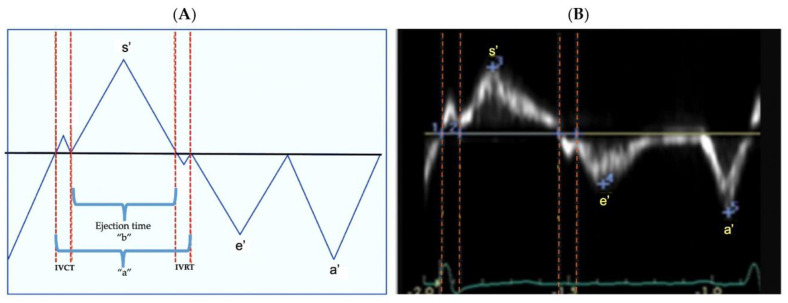
TDI derived myocardial velocities. Peak systolic velocity, s’, occurs during ejection time. The two diastolic velocities are e’ and a’ and correspond to myocardial movement during passive and active filling, respectively. Isovolumetric relaxation time (IVRT) is the time interval between the closure of the aortic valve and opening of the mitral valve occurring from the end of the s’ wave and the onset of the e’ wave. Isovolumetric contraction time (IVCT) is the time interval between the closure of the mitral valve and opening of the aortic valve occurring from the end of the a’ and onset of s’. Myocardial performance index is calculated from the formula (a–b/b). (**A**) Schematic representation. (**B**) TDI image.

**Figure 3 jcm-10-02211-f003:**
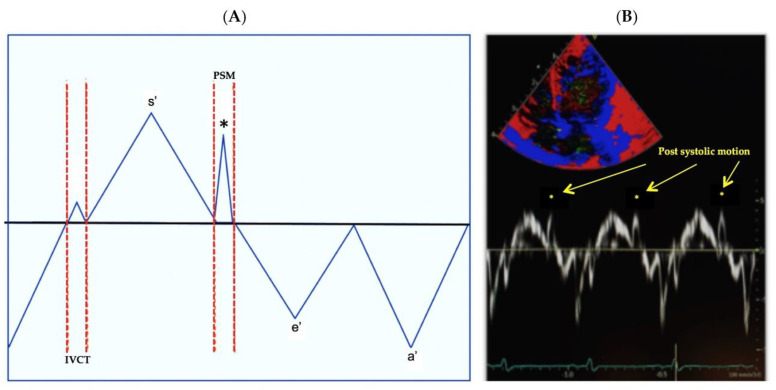
Post systolic motion (PSM) is denoted by the (*) and identified as a positive deflection on TDI occurring between the end of s’ and prior to e’. (**A**) Schematic representation. (**B**) TDI image.

**Figure 4 jcm-10-02211-f004:**
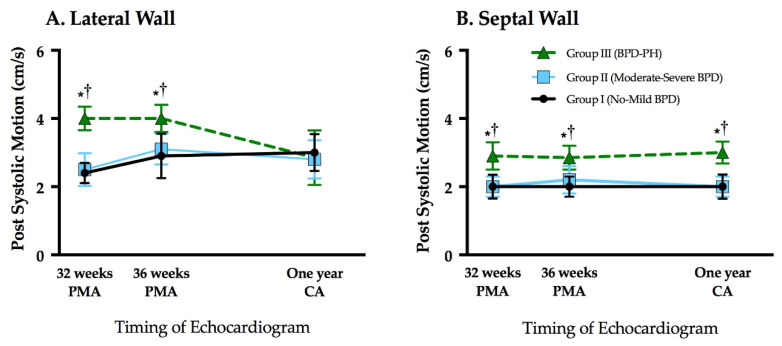
Post systolic motion (velocity) tracked over time in the (**A**) left ventricle and (**B**) interventricular septal wall between Group I, no-mild BPD (black dots), Group II, moderate-severe BPD (blue squares), and Group III, BPD-PH (green triangles). BPD, bronchopulmonary dysplasia; PH, pulmonary hypertension; PMA, postmenstrual age; CA, corrected age. * *p* < 0.05 Group I vs. Group III, comparisons adjusted for gestational age at birth, sex, and heart rate. ^†^
*p* < 0.05 Group II vs. Group III, comparisons adjusted for gestational age at birth, sex, and heart rate.

**Table 1 jcm-10-02211-t001:** Demographics and characteristics of preterm cohorts.

	Group I No-Mild BPD (*n* = 33)	Group II Mod-Sev BPD (*n* = 33)	Group III BPD-PH (*n* = 11)	F-Ratio *p* Value
Gestational age (weeks)	27 (26–28)	26 (25–27) ∞	26 (25–27) *	<0.01
Birth weight (grams)	950 (818–1070)	890 (670–985)	920 (860–1030)	0.58
Female, No. (%)	19 (58%)	17 (52%)	2 (19%) *	0.02
Infant Race, No. (%)				0.14
White	13 (39%)	15 (45%)	4 (36%)	
Black	20 (61%)	18 (55%)	7 (64%)	
Antenatal Steroids—No. (%)	29 (88%)	23 (70%)	8 (73%)	0.51
Surfactant therapy (Yes)	33 (100%)	33 (100%)	11 (100%)	>0.99
Cesarean section	24 (73%)	22 (67%)	4 (36%)	
Maternal complications, No. (%)				
Gestational Diabetes Mellitus	1 (3%)	3 (10%)	0	0.35
PROM	5 (15%)	2 (6%)	2 (18%)	0.61
Chorioamnionitis	3 (10%)	3 (10%)	1 (9%)	0.65
Pre-eclampsia	4 (12%)	5 (15%)	3 (27%)	0.78
Postnatal complications, No. (%)				
Necrotizing enterocolitis	2 (6%)	3 (9%)	1 (9%)	0.65
ROP threshold (>Stage 2)	9 (27%)	13 (39%)	4 (36%)	0.15
IVH (Grade 3 or 4)	5 (15%)	6 (18%)	6 (54%) *^,†^	0.02
Presence of PDA at 36 weeks	3 (9%)	4 (12%)	1 (9%)	0.23
Postnatal steroid use	3 (6%)	17 (51%) ∞	4 (36%) *	<0.01
Mechanical ventilation days	2 (1–5)	18 (2–30) ∞	18 (2–41) *	<0.01
Total oxygen days (NICU)	46 (28–76)	98 (88–115) ∞	97 (66–118) *	<0.01
Length of stay (NICU)	81 (69–101)	113 (91–120) ∞	101 (66–119) *	<0.01

Data are presented as median (interquartile range) or number (percentage). Mod-Sev, moderate-severe; BPD, Bronchopulmonary dysplasia; PH, pulmonary hypertension; PROM, premature rupture of membranes; ROP, retinopathy of prematurity; IVH, intraventricular hemorrhage; PDA, patent ductus arteriosus; NICU, neonatal intensive care unit. * *p* < 0.05 Group I vs. Group III, comparisons adjusted for gestational age at birth, sex, and heart rate. ∞ *p* < 0.05 Group I vs. Group II, comparisons adjusted for gestational age at birth, sex, and heart rate. ^†^
*p* < 0.05 Group II vs. Group III, comparisons adjusted for gestational age at birth, sex, and heart rate.

**Table 2 jcm-10-02211-t002:** Criteria for pulmonary hypertension diagnosis at 36 weeks postmenstrual age.

Patient	GA	BPD Severity	PH *	Designated Criteria for Diagnosis of PH						
				PDA Shunt **	PFO/ASD Shunt **	VSD Shunt **	RVSP∞ > 40 mmHg	RVSP/sBP > 0.5	RV Morphology Changes ^†^	Septal Wall Flattening
1	28	Mild	Yes	No	No	No	Yes	Yes	No	Moderate
2	27	Mild	Yes	No	No	No	Yes	Yes	No	None
3	25	Mild	Yes	No	No	No	Yes	Yes	No	Moderate
4	25	Mild	Yes	No	No	No	Yes	Yes	No	None
5	27	Moderate	Yes	No	No	No	Yes	Yes	No	None
6	24	Moderate	Yes	No	No	No	Yes	Yes	No	None
7	27	Moderate	Yes	No	No	No	Yes	Yes	No	None
8	27	Moderate	Yes	No	No	No	Yes	Yes	No	Moderate
9	25	Moderate	Yes	No	No	No	Yes	Yes	No	None
10	25	Severe	Yes	No	No	No	Yes	Yes	Yes	Moderate
11	26	Severe	Yes	No	No	No	Yes	Yes	No	Moderate

* PH, pulmonary hypertension; patients were classified with late onset PH at 36 weeks PMA if they had two or more of the designated criteria from Mourani et al. [24]. BPD, bronchopulmonary dysplasia (based on a modified definition of the 2001 National Institutes of Health BPD workshop) [22]. ** None of the 11 patients had bidirectional, or a right to left shunt pattern through the PDA, PFO/ASD, or VSD. Only one patient, #11, had a PDA (small) at 36 weeks PMA. All 11 patients had either a PFO or ASD (<1 mm). None of the 11 infants had a VSD at 36 weeks PMA. GA, gestational age; RVSP, right ventricle systolic pressure; sBP, systolic blood pressure; PDA, patent ductus arteriosus; PFO, patent foramen ovale; ASD, atrial septal defect; VSD, ventricular septal defect. ∞ RVSP was based on the presence of the Doppler velocity tricuspid regurgitate jet (modified Bernoulli equation). An interpretable DVTR was present in 14.5% (11/77) of patients at 36 weeks postmenstrual age. Only 5/77 (7%) had any degree of septal wall flattening at 36 weeks PMA. ^†^ RV morphology changes were defined as RV hypertrophy or dilatation. Patient #10 had RV dilatation.

**Table 3 jcm-10-02211-t003:** LV tissue Doppler indices at the lateral side of the mitral valve annulus.

	Group I No-Mild BPD (*n* = 33)	Group II Mod-Sev BPD (*n* = 33)	Group III BPD-PH (*n* = 11)	F-Ratio *p* Value
32 weeks PMA				
LV e’ (cm/s)	4.0 (4.0–5.0)	4.0 (3.0–4.0)	3.0 (3.0–4.0)	0.19
LV a’ (cm/s)	8.0 (7.0–9.0)	7.0 (6.0–9.0)	9.0 (8.0–10.0)	0.23
LV s’ (cm/s)	5.0 (4.0–5.0)	5.0 (4.0–5.0)	5.0 (4.0–5.0)	0.98
LV e’/a’	0.57 (0.42–0.68)	0.44 (0.37–0.62)	0.46 (0.44–0.62)	0.45
LV IVRT (ms)	48 (45–58)	52 (45–55)	68 (60–69) *^,†^	<0.01
LV MPI	0.49 (0.43–0.58)	0.52 (0.44–0.65)	0.51 (0.43–0.57)	0.32
LV PSM (cm/s)	2.5 (2.0–3.0)	2.5 (2.0–3.0)	4.0 (3.0–5.0) *^,†^	<0.01
36 weeks PMA				
LV e’ (cm/s)	4.0 (4.0–6.0)	4.0 (3.0–4.0)	3.0 (3.0–5.0)	0.14
LV a’ (cm/s)	8.0 (7.0–10.0)	9.0 (7.0–10.0)	7.0 (5.0 –10.0)	0.56
LV s’ (cm/s)	5.0 (4.0–5.0)	5.0 (4.0–6.0)	5.0 (4.0 –5.0)	0.80
LV e’/a’	0.57 (0.50–0.67)	0.46 (0.34–0.50)	0.50 (0.40–0.60)	0.67
LV IVRT (ms)	52 (47–58)	54 (47–61)	67 (61–74)	<0.01
LV MPI	0.54 (0.45–0.64)	0.56 (0.49–0.62)	0.52 (0.50–0.64)	0.42
LV PSM (cm/s)	3.0 (2.0–3.0)	3.0 (2.0–4.0)	4.0 (2.8–4.5) *^,†^	<0.01
One year CA				
LV e’ (cm/s)	12.0 (11.0–13.0)	10.0 (8.0–13.0)	12.0 (9.0–15.0))	0.4
LV a’ (cm/s)	7.0 (6.0–8.0)	7.0 (6.0–8.0)	8.0 (6.0–0.0)	0.93
LV s’ (cm/s)	7.0 (6.0–8.0)	7.0 (6.0–8.0)	7.0 (6.0–8.0)	0.20
LV e’/a’	1.44 (1.31–1.85)	1.44 (1.31–1.85)	1.38 (1.33–1.88)	0.67
LV IVRT (ms)	50 (45–57)	49 (44–53)	52 (48–57)	0.43
LV MPI	0.42 (0.38–0.46)	0.43 (0.39–0.48)	0.43 (0.39–0.51)	0.64
LV PSM (cm/s)	3.0 (2.0–4.0)	3.0 (2.0–3.0)	3.0 (2.0–3.0)	0.56

Data are presented as median (interquartile range) or number (percentage). Mod-Sev, moderate-severe; LV, left ventricle; IVRT, isovolumetric relaxation time; MPI, myocardial performance index; PSM, post systolic motion; PMA, post menstrual age; CA, corrected age. * *p* < 0.05 Group I vs. Group III, comparisons adjusted for gestational age at birth, sex, and heart rate. ^†^
*p* < 0.05 Group II vs. Group III, comparisons adjusted for gestational age at birth, sex, and heart rate.

**Table 4 jcm-10-02211-t004:** LV tissue Doppler indices at the septal side of the mitral valve annulus.

	Group I No-Mild BPD (*n* = 33)	Group II Mod-Sev BPD (*n* = 33)	Group III BPD-PH (*n* = 11)	F-Ratio *p* Value
32 weeks PMA				
Septal e’ (cm/s)	4.0 (4.0–5.0)	4.0 (3.0–4.0)	5.0 (4.0–5.0)	0.40
Septal a’ (cm/s)	7.0 (6.0–8.0)	6.0 (6.0–7.0)	8.0 (6.0–8.0)	0.42
Septal s’ (cm/s)	4.0 (4.0–5.0)	4.0 (4.0–5.0)	4.0 (4.0–4.0)	0.27
Septal e’/a’	0.56 (0.50–0.68)	0.57 (0.50–0.67)	0.65 (0.63–0.70)	0.34
Septal IVRT (ms)	45 (41–53)	46 (40–53)	52 (43–61) *^,†^	<0.01
Septal MPI	0.43 (0.39–0.50)	0.46 (0.41–0.50)	0.46 (0.37–0.53)	0.45
Septal PSM (cm/s)	2.0 (2.0–2.0)	2.0 (2.0–2.0)	2.5 (2.0–3.0) *^,†^	<0.01
36 weeks PMA				
Septal e’ (cm/s)	5.0 (3.0–5.0)	4.0 (3.0–4.0)	4.0 (4.0–5.0)	0.34
Septal a’ (cm/s)	7.0 (6.0–8.0)	7.0 (5.0–7.0)	8.0 (7.0–9.0)	0.65
Septal s’ (cm/s)	5.0 (4.0–5.0)	5.0 (4.0–5.0)	5.0 (4.0–6.0)	0.39
Septal e’/a’	0.63 (0.45–0.71)	0.57 (0.43–0.71)	0.56 (0.49–0.70)	0.67
Septal IVRT (ms)	46 (40–53)	46 (41–54)	52 (45–58) *^,†^	<0.01
Septal MPI	0.44 (0.41–0.49)	0.44 (0.40–0.51)	0.44 (0.37–0.51)	0.74
Septal PSM (cm/s)	2.0 (2.0–3.0)	2.0 (2.0–2.0)	2.5 (2.0–3.3) *^,†^	<0.01
One year CA				
Septal e’ (cm/s)	10.0 (10.0–11.0)	10.0 (9.0–11.0)	10.0 (8.0–11.0)	0.14
Septal a’ (cm/s)	8.0 (6.0–9.0)	9.0 (7.0–10.0)	7.0 (6.0–9.0)	0.24
Septal s’ (cm/s)	7.0 (6.0–8.0)	7.0 (6.0–8.0)	7.0 (6.0–7.0)	0.29
Septal e’/a’	1.4 (1.13–1.74)	1.26 (0.88–1.67)	1.25 (1.03– 1.86)	0.67
Septal IVRT (ms)	51 (46–54)	50 (44–54)	58 (44–64) *^,†^	<0.01
Septal MPI	0.40 (0.35–0.44)	0.40 (0.37–0.43)	0.38 (0.33–0.45)	0.50
Septal PSM (cm/s)	2.0 (2.0–3.0)	2.0 (2.0–3.0)	3.0 (2.0–3.0) *^,†^	<0.01

Data are presented as median (interquartile range) or number (percentage). Mod-Sev, moderate-severe; LV, left ventricle; IVRT, isovolumetric relaxation time; MPI, myocardial performance index; PSM, post systolic motion; PMA, post menstrual age; CA, corrected age. * *p* < 0.05 Group I vs. Group III, comparisons adjusted for gestational age at birth, sex, and heart rate. ^†^
*p* < 0.05 Group II vs. Group III, comparisons adjusted for gestational age at birth, sex, and heart rate.

**Table 5 jcm-10-02211-t005:** Comparison of LV tissue Doppler indices between term and preterm individuals at one year of age.

	Group I No-Mild BPD (*n* = 33)	Group II Mod-Sev BPD (*n* = 33)	Group III BPD-PH (*n* = 11)	Term Control (*n* = 50)	F-Ratio *p* Value
LV medial (Septal) wall					
Septal e’ (cm/s)	10.0 (10.0–11.0)	10.0 (9.0–11.0)	10.0 (8.0–11.0)	11.0 (9.0–12.0)	0.11
Septal a’ (cm/s)	8.0 (6.0–9.0)	9.0 (7.0–10.0)	7.0 (6.0–9.0)	7.4 (6.3–9.5)	0.29
Septal s’ (cm/s)	7.0 (6.0–8.0)	7.0 (6.0–8.0)	7.0 (6.0–7.0)	7.9 (6.5–9.0)	0.24
Septal e’/a’	1.4 (1.13–1.74)	1.26 (0.88–1.67)	1.25 (1.03–1.86)	1.40 (1.23–1.66)	0.27
Septal IVRT (ms)	51 (46–54)	50 (44–54)	58 (44–64) *^,†^	46 (40–54) ^c^	<0.01
Septal MPI	0.40 (0.35–0.44)	0.40 (0.37–0.43)	0.38 (0.33–0.45)	0.35 (0.31–0.43)	0.30
Septal PSM (cm/s)	2.0 (2.0–3.0)	2.0 (2.0–3.0)	3.0 (2.0–3.0) *^†^	NR	NA
LV lateral wall					
LV e’ (cm/s)	12.0 (11.0–13.0)	10.0 (8.0–13.0)	12.0 (9.0–15.0)	12.9 (11.0–15.0)	<0.01
LV a’ (cm/s)	7.0 (6.0–8.0)	7.0 (6.0–8.0)	8.0 (6.0–0.0)	9.0 (7.0–10.0)	0.23
LV s’ (cm/s)	7.0 (6.0–8.0)	7.0 (6.0–8.0)	7.0 (6.0–8.0)	8.5 (7.0–10.0) ^a,b,c^	<0.01
LV e’/a’	1.44 (1.31–1.85)	1.44 (1.31–1.85)	1.38 (1.33–1.88)	1.28 (113–1.68) ^a,b,c^	<0.01
LV IVRT (ms)	50 (45–57)	49 (44–53)	52 (48–57)	44 (38–51) ^a,b,c^	<0.01
LV MPI	0.42 (0.38–0.46)	0.43 (0.39–0.48)	0.43 (0.39–0.51)	0.36 (0.31–0.49) ^a,b,c^	<0.01
LV PSM (cm/s)	3.0 (2.0–4.0)	3.0 (2.0–3.0)	3.0 (2.0–3.0)	NR	NA

Data are presented as median (interquartile range) or number (percentage). Mod-Sev, moderate-severe; LV, left ventricle; IVRT, isovolumetric relaxation time; MPI, myocardial performance index; PSM, post systolic motion; NR, not recorded. ^a^
*p* < 0.05 Group I vs. Term, ^b^
*p* < 0.05 Group II vs. Term, ^c^
*p* < 0.05 Group III vs. Term, * *p* < 0.05 Group I vs. Group III, comparisons adjusted for gestational age at birth, sex, and heart rate. ^†^
*p* < 0.05 Group II vs. Group III, comparisons adjusted for gestational age at birth, sex, and heart rate.

**Table 6 jcm-10-02211-t006:** Current literature on tissue Doppler velocities at one year of age.

	Preterm		Term			
	Torres *	Poon 2019 [43] **	Torres	Alp 2012 [46]	Choi 2016 [45]	Poon 2019 [43]
LV a’ (cm/s)	11.8 (1.7)	10.8 (1.2)	12.8 (1.2)	13.3 (0.9)	11.3 (1.5)	11.7 (2.1)
LV a’ (cm/s)	7.2 (1.6)	3.6 (1.0)	8.2 (1.3)	11.1 (1.2)	6.2 (1.3)	3.7 (1.6)
LV s’ (cm/s)	6.5 (0.8)	4.7 (1.0)	8.4 (0.9)	9.9 (0.8)	7.0 (0.79)	4.8 (1.0)
LV e’/a’	1.6 (0.3)	NR	1.3 (0.2)	1.2 (0.1)	NR	1.4 (0.2)
LV IVRT (s)	50.2 (7.4)	NR	45 (3)	NR	NR	NR
LV PSM (cm/s)	3.0 (1.0)	NR	NR	NR	NR	NR

Data presented as mean (standard deviation); NR, not recorded. * Preterm born individuals with no lung or pulmonary vascular disease ** Preterm control individuals without initial respiratory distress syndrome.

## Data Availability

The data that support the findings of this study are available on request from the corresponding author. The data are not publicly available due to privacy restrictions.

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
