# Peer review of "Left Ventricle Phenotyping Utilizing Tissue Doppler Imaging in Premature Infants with Varying Severity of Bronchopulmonary Dysplasia"

_jcm, 2021, doi:10.3390/jcm10102211_

Round 1
Reviewer 1 Report
Please, see attached file

Reviewer 2 Report
Please see attachment

Round 2
Reviewer 1 Report
The manuscript has been revised satisfactorily by authors. The new title is well set and no longer misleading.
There are just a few remaining questions/corrections to be addressed.
-Table 2 (new) is interesting. The data shows that diagnosing PH at 36 weeks PMA according to Mourani criteria may be of minor significance, especially in children without severe BPD, as none had PH at one year CA. More likely, the findings of PH at 36 weeks PMA in these children may be a result of late reduction of pulmonary resistance (normal transition) in these preterm infants. I agree on your statement in section 4.5, about children with more severe phenotypes of PH need to be studied to be able to interpret the clinical significance of PSM measurements.
- Comment to point 20, section 4.4: I agree on PSM being able to combine elements from longitudinal, electrical and rotational components, but to my knowledge, IVRT is affected not only by longitudinal but also by electrical components?
-Section 3.3 is interesting, and the individual hemodynamics and interpretation of PDAs in preterms will continue to elude neonatologists.
Specific comments:
- Thank you for changing medial to septal wall consequently. However, the title of Table 4 is still to be corrected.
- Legend of Fig 4: IVRT and MPI are inaccurately defined, as there are no data of these variables included in figure. Please, correct.
- Table 5 (new) is redundant as you already revised the text accurately on reproducibility in section 3.2.
Author Response
2nd Response to Reviewer 1 Comments
Point 1: The manuscript has been revised satisfactorily by authors. The new title is well set and no longer misleading.
Response to Point 1: Thank you
There are just a few remaining questions/corrections to be addressed.
Point 2: Table 2 (new) is interesting. The data shows that diagnosing PH at 36 weeks PMA according to Mourani criteria may be of minor significance, especially in children without severe BPD, as none had PH at one year CA. More likely, the findings of PH at 36 weeks PMA in these children may be a result of late reduction of pulmonary resistance (normal transition) in these preterm infants. I agree on your statement in section 4.5, about children with more severe phenotypes of PH need to be studied to be able to interpret the clinical significance of PSM measurements.
Response to point 2: Thank you for this observation. We agree that the echocardiographic findings of late PH may not have clinical significance and may be due to the late reduction of PVR. Previous evidence has also shown a persistently elevated PVR through one year of age in these children.
Point 3: Comment to point 20, section 4.4: I agree on PSM being able to combine elements from longitudinal, electrical and rotational components, but to my knowledge, IVRT is affected not only by longitudinal but also by electrical components?
Response to Point 3: Thank you for this comment. IVRT reflects the time interval between aortic valve closure and mitral valve opening, and is both load and age dependent. For example, when the LV relaxation is impaired, the opening of the mitral valve will be delayed and will prolong the IVRT. Ageing is also associated with prolongation and a reduction in early diastolic filling of the LV. Similar patterns have also been seen in the preterm population. We agree with the reviewer that IVRT could also be affected by electrical activity, such as delayed repolarization noted in cardiomyopathies and in patients with bundle branch block where IVRT may be prolonged. Prolongation of IVRT in these conditions reflect abnormal synchronous LV relaxation due to alterations in electrical activity but these conditions also exhibit PSM due to abnormal rotational/torsional mechanics. We have added language to section 4.4 Clinical significance to reflect this important point on page 13.
Point 4: Section 3.3 is interesting, and the individual hemodynamics and interpretation of PDAs in preterm will continue to elude neonatologists.
Response to Point 3: Thank you. We agree
Specific comments:
Point 5: Thank you for changing medial to septal wall consequently. However, the title of Table 4 is still to be corrected.
Response to Point 5: Thank you. We have changed the title of Table 4.
Point 6: Legend of Fig 4: IVRT and MPI are inaccurately defined, as there are no data of these variables included in figure. Please, correct.
Response to Point 6: Thank you for this observation. We have adjusted accordingly and removed IVRT and MPI.
Point 7: Table 5 (new) is redundant as you already revised the text accurately on reproducibility in section 3.2.
Response to Point 7: Table has been removed.